# Supplementing sleep actigraphy with button pressing while awake

**Marius Keller** [1,2]*, **Walton T. Roth** [3,4], **Katja Petrowski** [1,5]

**1** Medical Psychology and Medical Sociology, University Medical Center of the Johannes Gutenberg University Mainz, Mainz, Germany, **2** Department of Neurology, Elblandklinikum Meissen, Meissen, Germany, **3** Department of Psychiatry and Behavioral Sciences, Stanford University School of Medicine, Stanford, California, United States of America, **4** War Related Illness and Injury Center, VA Palo Alto Health Care System, Palo Alto, California, United States of America, **5** Department of Psychotherapy and Psychosomatic Medicine, Dresden University of Technology, Medical School, Dresden, Germany

* keller.marius@outlook.com

## Abstract

### Objective/background

Wrist-worn sleep actigraphs are limited for evaluating sleep, especially in sleepers who lie awake in bed without moving for extended periods. Sleep logs depend on the accuracy of perceiving and remembering times of being awake. Here we evaluated pressing an event-marker button while lying awake under two conditions: self-initiated pressing every 5 to 10 minutes or pressing when signaled every 5 minutes by a vibration pulse from a wristband. We evaluated the two conditions for acceptability and their concordance with actigraphically scored sleep.

### Participants and methods

Twenty-nine adults wore actigraphs on six nights. On nights 1 and 4, they pressed the marker to a vibration signal, and on nights 2 and 5, they self-initiated presses without any signal. On nights 3 and 6, they were told not to press the marker. Every morning they filled out a sleep log about how they had slept.

### Results

The vibration band was unacceptable to 42% of the participants, who judged it too disturbing to their sleep. Self-initiated pressing was acceptable to all, although it reduced log reported sleep depth compared to a no pressing condition. Estimations of sleep onset latency were considerably longer by button pressing than by actigraphy. Agreement of epoch-by-epoch sleep scoring by actigraphy and by button pressing was poor (kappa = 0.23) for self-initiated pressing and moderate (kappa = 0.46) for pressing in response to a vibration.

### Conclusions

Self-initiated button pressing to indicate being awake while lying in bed is acceptable to many, interferes little with sleep, and adds substantially to the information given by actigraphy.

**Data Availability Statement:** All relevant data are within the paper and its Supporting Information files.

**Funding:** The authors received no specific funding for this work.

**Competing interests:** The authors have declared
that no competing interests exist.

## Introduction

Actigraphy (ACT), which is based on measuring body movement, has been used to evaluate sleep for more than 50 years. Its advantage over the gold standard, EEG polysomnography (PSG), is simplicity, low cost, low subject burden, and ability to record for weeks in the subject's own environment. The claim for popular wrist devices to evaluate sleep quality depends principally on actigraphy, i.e., on their containing an accelerometer sensing body movement. Some commercially available devices lack documentation on scoring, and many have been proven to have low accuracy [1], especially in certain populations, such as patients with obstructive sleep apnea [2]. Studies in which PSG and research-based actigraphy were recorded simultaneously have found that actigraphic scoring usually underestimates Sleep Onset Latency (SOL) and Wake after Sleep Onset (WASO) compared to PSG, especially in sleepers who lie awake without moving for extended periods [3], which is common in people suffering insomnia [4]. Actigraphic sensitivity for PSG-scored sleep can be high (up to 97%), but specificity is low (43%) because actigraphy misclassifies PSG wake periods as sleep [5]. Such findings led Ancoli-Israel, Martin, et al. [6] to recommend supplementing actigraphy data with other sources such as sleep logs filled out the next morning.

Another way to supplement actigraphy data would be to combine them with behavioral measures of being awake. In 1939, SOL was estimated by observing when a hand-held object was dropped [7, 8]. Other investigators used a "dead man switch", which closes a circuit when no longer pressed [9, 10], or an auditory signal detection task [11]. Casagrande, De Gennaro, et al. [12] compared self-initiated responding and response to signals as indications of being awake. They concluded that a finger-tapping task interfered with falling asleep less than responding to auditory stimuli every 6 seconds. Haptic stimuli have also been used as signals [13]. In general, signals need to be set to be clearly perceptible during waking but not so intense as to wake up the sleeper, which is complicated because detection thresholds rise with drowsiness and vary in different sleep phases. An early study found that thresholds to tactile stimuli rose to 100 times the pre-sleep level after sleep onset [8]. Some ways of supplementing information may turn out to be less accurate compared to other methods of assessing sleep or less acceptable to subjects. Thurman, Wasylyshyn, et al. [14] found that compared to actigraphy both accuracy and compliance with filling out sleep logs changed over consecutive weeks.

In the current study, we focused on behavioral measures of sleep, knowing that they may define sleep differently from PSG [11]. We compared a haptic signal detection task with a task in which subjects are instructed to self-initiate event-marker button presses while lying in bed. We were interested both in how well these two task conditions corresponded to actigraphic estimates of sleep, and in how acceptable they were as not interfering with the person's usual sleep quality.

## Methods

### Participants

Thirty potential subjects were recruited in the USA (21) and Germany (9). One decided not to participate after the experiment was explained, leaving 29 subjects. Any adults were eligible, regardless of health status or whether they took medications affecting sleep. Ages ranged from 19 to 76 (mean = 36.1, SD = 14.5). Nine subjects were men and 20 women. Pittsburgh Sleep Quality Inventory (PSQI) [15] total scores ranged from 2 to 13 (mean = 5.77, SD = 2.8). One subject used a continuous positive airway pressure mask and one took a hypnotic nightly. Upon completion of 6 nights of testing, American participants were eligible to receive $100.

The study was approved by the Stanford and VA Palo Alto human research committees and the Ethikkommission of the Dresden University of Technology. Written informed consent was obtained from all individual participants included in the study.

## Devices

Two wrist-worn devices were used, an actigraph and a vibration wristband. The actigraph was the Motionlogger® (Ambulatory Monitoring Inc., Ardsley, USA), worn on the non-dominant wrist and set to record data in 10-second epochs. This actigraph has an event marker button, which when pressed, gives a feedback tone and records the time. Its built-in memory stored 6 days of recording for analysis by Action-W Version 2.7.3045 software (Ambulatory Monitoring Inc., Ardsley, USA) (AW2.7). The vibration wristband was constructed from a wrist sweatband that enclosed a programmable microcontroller and vibration motor powered by disc batteries, which delivered 2-second vibrations every 5 minutes. The intensity of the vibrations, the same for every subject, was set in pilot tests to be felt as weak but easily perceptible. For the vibration motor used, the vibration amplitude at 3.0V is specified to be 1 G at a frequency of 183Hz ± 50Hz, we ran the motor at 0.8 V—this resulted in vibrations of 80 Hz at an approximate amplitude of 0.3 G. For detailed technical information on the wristband setup, see S2 File.

## Procedure

After signing consent forms, subjects received oral and written study instructions, the PSQI to be answered for the past month, and six sleep logs. The recordings were made at home. There were no requirements regarding sleep/wake schedule or daytime activities. Subjects were asked to wear the actigraph for six days and nights. We explained the three conditions, and that each condition was to be undergone twice in a predefined order. Each morning they were to fill out a sleep log about how they had slept last night. Although subjects usually wore the actigraph on consecutive nights, a few subjects skipped nights, continuing in their assigned condition on the following night: two skipped one night (one after night 2 and the other after night 5), and two skipped two nights (one after nights 2 and 3 and the other for 2 nights after night 3).

**Pressing in response to a vibration pulse (RP).**   On RP nights (night 1 and 4), subjects were to put the vibration band on their dominant wrist just before going to bed intending to sleep, and to take it off after finally getting out of bed. They were to press the marker once when going to bed and once when getting out of bed, as well as whenever they felt a vibration. Thus, they would press while initially waiting to fall asleep, while awake during the sleeping period, and after awakening in the morning before getting out of bed. If the vibration band made it too difficult or impossible to sleep, they were told they could remove it.

**Self-initiated pressing (SP).**   On SP nights (night 2 and 5), subjects wore only the actigraph. Markers were to be pressed when going to bed with the intention to sleep and when getting out of bed as on RP nights. But instead of being reminded to press the marker by a vibratory signal, lying awake was to be signaled by pressing the marker self-initiated once every 5 to 10 minutes. Instructions emphasized that this time interval was only a rough guide, not to be checked by looking at a clock. The 5- to 10-minute interval was intended to reduce worry about exactly estimating the time. If participants felt the SP instructions were a significant barrier to falling asleep, they could discontinue pressing, documenting that in the sleep log.

**No Pressing (NP).**   On NP nights (night 3 and 6), the actigraph was worn but the marker was never to be pressed.

## Sleep logs

Sleep logs were to be filled out every morning soon after getting out of bed. We used a modified version of the consensus sleep diary (CSD-core) [16] with ordinal scales for assessing SOL, number of wake episodes (NWE) and WASO. Our log was simplified by not asking for time of getting in and out of bed since they were given by marker presses or actigraphy. We extended the log with questions assessing three, rather than one, sleep parameter using ordinal scales with four levels: Overall Quality (poor/medium/good/excellent), Length (too short/a little too short/good/too long), and Depth (too light/a little too light/good/too deep). In addition, subjects were asked about their compliance with instructions: for RP they were asked whether they had worn the vibration band for the whole night, parts of the night, or not at all; for SP, how often they pressed the marker while awake (seldom/usually/ always). For both RP and SP they were asked, "Did pushing the button/vibrations make it harder to fall asleep?" (no/somewhat/much).

## Data analysis

### Sleep parameters

For RP and SP, we calculated Time in Bed (TIB) from markers pressed when lying down intending to sleep and when getting up in the morning. For NP, participants were asked not to press markers, although some did. In the absence of markers, TIB was set using the AW2.7 Auto Set Down Interval function. If by inspection the resulting start or end time was incongruent with activity patterns, we selected epochs with an activity count of zero that bordered on previous or following epochs with levels of daytime activity [17].

Some evidence suggests that data collected in Proportional Integrating Mode (PIM) produce more reliable results than that collected in Zero Crossing Mode (ZCM) for actigraphic sleep scoring [18–20], and so we used the UCSD-PIM scoring algorithm described in Jean-Louis, Kripke [5]. For RP and SP marker sleep scoring, an epoch (i.e. a 10-second time interval) containing a marker press was scored as wake. If two epochs scored wake were less than 10 minutes and 10 seconds apart, intervening epochs were also scored as wake. This is twice the interval between vibrations (in RP) plus 10 seconds for response latency.

The following sleep measures were calculated from activity and markers: SOL, WASO, NWE, Wake Time after Sleep Offset (WASF), and Sleep Percentage (Sleep%). Table 1 gives definitions of these parameters. Variables based on actigraphy were calculated by Action-W Version 2.7.3045 software, and those based on markers using a script written in Matlab 2015a (The MathWorks, Inc., Natick, Massachusetts, United States). For SOL by actigraphy we used a 20-minute criterion: "the beginning of the first period containing 20 minutes of actigraph-identified sleep with no more than one minute of wake intervening" [21]. This criterion was proposed by Cole, Kripke, et al. [22] to improve correlation of actigraphic sleep onset with PSG, and is the default setting in Action-W2.7.

### Statistics

Statistical tests were performed using IBM SPSS Statistics for Windows, version 22 (IBM Corp., Armonk, N.Y., USA). RP nights were excluded from analyses that included the RP condition if the subject did not wear the vibration band for both entire nights. In SP, all records were considered valid. For comparison of log-reported sleep parameters, we used non-parametric Wilcoxon signed-rank tests and Friedman tests. For post-hoc analyses of two conditions, Bonferroni corrections were applied. *Log-reported* sleep parameters were compared for the two repetitions of each condition. Friedman tests were applied across all three

**Table 1. Definition of sleep parameters.**

| Sleep parameter | Actigraphy as per AW2.7 | Marker-based |
|---|---|---|
| TIB | Time the subject spent in bed with the intention to sleep. | |
| Sleep Onset | The clock time of the first epoch[a] scored as sleep of the first 20-minute block containing not more than one minute of 'wake'. | The clock time of the last marker in the first block of consecutive[b] markers. |
| SOL | Time until Sleep Onset after beginning of TIB | |
| Sleep Offset | The clock time associated with the last epoch[a] scored as 'sleep' prior to the end of Time in Bed. | The clock time of the first marker in the last block of consecutive[a] markers. |
| WASO | The sum of all minutes scored as wake between Sleep Onset and Sleep Offset | The overall duration of consecutive[a] marker presses between Sleep Onset and Sleep Offset. Single marker presses add 10 seconds. |
| NWE | The number of alternations between 'sleep' and 'wake' during TIB. | |
| WASF | Time following Sleep Offset until end of TIB | |
| Sleep% | Proportion of time scored sleep during TIB | |

[a] An epoch was defined as a 10-second time interval.

[b] Markers were considered consecutive when less than 10 minutes and 10 seconds apart.

conditions for subjects who had worn the vibration band for two entire nights. For all subjects, log reported sleep parameters in SP vs. NP were compared.

RP and SP *actigraphic* sleep parameters were compared to those in NP nights. Non-parametric tests were used for comparisons with actigraphically measured sleep parameters: SOL, WASO, NWE, WASF and Sleep%, since Shapiro-Wilk tests and inspection of normal QQ-plots indicated that these parameters were not normally distributed. RP and SP *actigraphic* sleep parameters were compared to those provided by markers on the same night. Paired-sample Wilcoxon signed-rank tests and Spearman correlation analyses were done for SOL, WASO, NWE, WASF and Sleep%. Bland-Altman analyses were done for SOL and Sleep%.

## Epoch-by-epoch agreement

Event markers and activity are recorded synchronously by the Motionlogger. Time epochs were scored separately by markers and by activity as either sleep or wake. An error matrix was created comparing marker scoring from each pressing condition to actigraphy. Inter-condition agreement was quantified by Cohen's kappa (K). Since sleep epochs are more prevalent than wake, K may underestimate agreement [23]. Thus, K's adjusted for prevalence and bias (PABAK) were also calculated [24]. PABAK has been used in more recent studies comparing sleep monitoring devices, e.g., [25] and [26], because it weights sleep and wake epochs identically by equalizing expected occurrences to 0.5 [24]. Following Landis and Koch [27], we considered kappa values of 0.0–0.2 to show slight agreement, 0.21–0.40 fair, 0.41–0.6 moderate, 0.61–0.80 substantial, and >0.80 almost prefect agreement.

## Results

Twenty-nine participants were included in the study. Two participants quit prematurely. One subject's records were missing after the first two nights. For two more subjects, records were missing or the actigraph failed to record. Overall, 75% (131/174) of the nights resulted in valid records, 53% (31/58) for RP, 88% (51/58) for SP and 84% (49/58) for NP.

## Acceptability

**General.** Wearing the Motionlogger at night was acceptable to everyone. One subject felt it irritated the skin on one wrist and shifted it to the other wrist. There were no significant differences in sleep log-assessed Sleep Quality, Depth, or Length between the first and second repetition for any condition.

**Pressing in response to a vibration pulse (RP).** 17 of 29 subjects (59%) accepted the vibration band for two full nights. Four did not put on the band for the second night, and one tried to wear it on her ankle but quit the study after that one night. Seven subjects wore it on both nights but took it off at least once during a night. One subject never tried to wear the band. A Fisher's Exact Test yielded acceptance for RP to be significantly lower than for SP, which was accepted by all subjects (p<0.001). One valid RP record was entirely lost due to an actigraph failure, and on one RP night, the subject pushed the wrong button and no markers were recorded.

Individuals who did not accept the band on both nights (n = 12) were not different in age (t (27) = -0.11, p = 0.913), sex ($\chi^2$(1,29) = 0.35, p = 0.55), or PSQI total score (t(27) = 0.12, p = 0.91) from those who fully accepted it (n = 17). Exploratory analyses did not find any PSQI subscale or actigraphic measure that significantly differed between these two groups.

A Friedman Test found no significant differences between the three conditions among subjects who had accepted the band on both nights (n = 17) for log-reported Quality or Length, but Depth was significantly lower ($\chi^2$(2,33) = 8.91, p = 0.012). Post-hoc Related Samples Wilcoxon signed-rank tests showed Depth to be significantly lower for both RP and SP compared to NP (RP vs. NP: Z = -2.49, adj. p = 0.04, SP vs. NP: Z = -2.53, adj. p = 0.03), but not for RP vs. SP: (Z = -0.63, p = 0.52). Fig 1 shows responses on the sleep log for sleep depth.

**Self-initiated pressing (SP).** No one refused the SP condition. On 88% of the nights, subjects reported to have pressed the button upon waking and to have continued pressing while

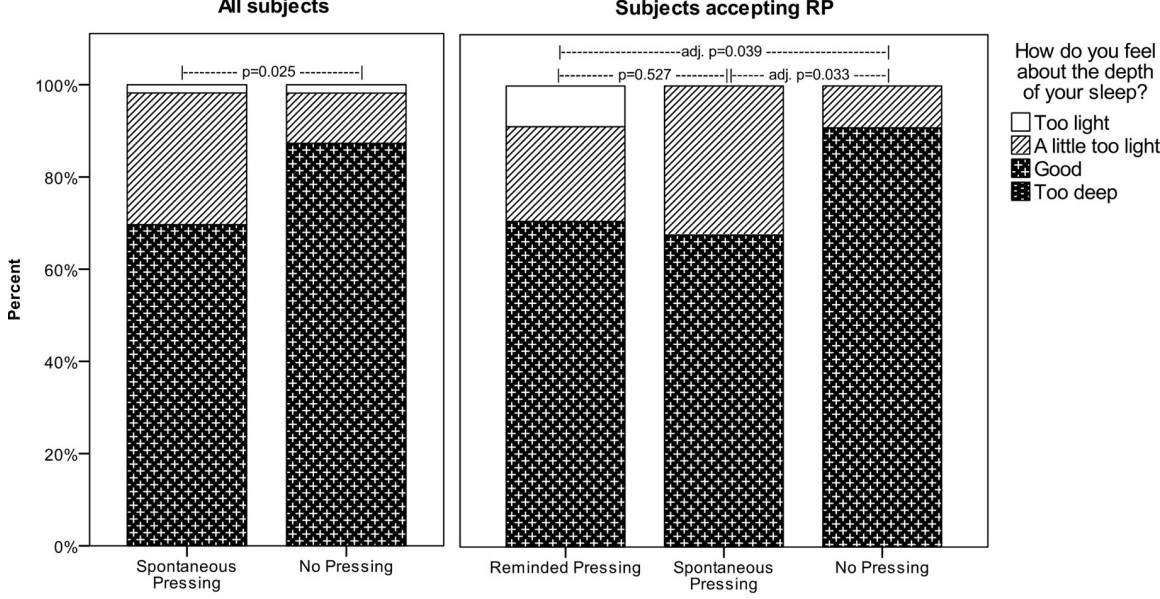

**Fig 1. Log-reported sleep depth under different conditions.** Left: Log-reported depth for all subjects under SP. P-value for Wilcoxon signed-rank test p = 0.025. Right: Log-reported depth for subjects who accepted RP under the three conditions. p-value across all groups in Friedman test p = 0.012. p-values for comparison of pairs of groups for Bonferroni adjusted Wilcoxon signed-rank tests are shown in the figure. SP = Self-initiated pressing, RP = Pressing in response to vibration pulse.

awake 'usually' or 'always'. On 12% of the nights, 'seldom' was reported. One subject failed to answer how much pressing the button impacted his sleep. No subject reported 'much', 13 reported 'somewhat' on 1 or 2 nights, and 14 reported 'no' on both nights.

Individuals who checked 'somewhat' for one or two nights did not significantly differ in age (t-test), sex (Fisher's Exact Test), or PSQI Total Score (t-test) from those who always checked 'no'.

Participants did not rate Quality or Length different between SP and NP (Wilcoxon signed-ranks test), but Depth was rated lower for SP than for NP (mdn = 'good' vs. mdn = 'good', Z = -2.24, p = 0.025).

## Actigraphic measures

No statistically significant differences were found between the two repetitions for any of the actigraphic or marker calculated sleep parameters. For this reason, sleep parameters from the repetitions were entered together into statistical analyses based on nights rather than on individual subjects. Table 2 gives the median and interquartile ranges (since the data were non-normal) for nights in the three conditions (non-shaded cells). Mann-Whitney-U-Tests for comparisons of *actigraphic data* in RP and SP versus NP found no differences for WASO, NWE, WASF, or Sleep% but SOL was significantly higher than in NP. Higher SOL *per actigraphy* in both conditions involving button presses (RP and SP) could either be explained by an artifactual effect of movement on the actigraph, which houses the marker button, when repeatedly pressed, or be explained by actual prolongation of the time to fall asleep (SOL). To control for this, further studies with simultaneous PSG recording would be necessary. Between RP and SP nights, no differences in any actigraphically measured sleep variable were found.

## Marker measures

Marker measures were obtained under RP and SP condition. The total number of markers pressed was significantly higher in RP than SP, while NWE was equal as is shown in the

**Table 2. Sleep measures scored by actigraphy and markers.**

| | RP (n = 31) | | | SP (n = 51) | | | NP (n = 49) | ACT | | | M |
|---|---|---|---|---|---|---|---|---|---|---|---|
| Measure | ACT | M | ACT vs. M | ACT | M | ACT vs. M | ACT | RP vs. NP | SP vs. NP | RP vs. SP | RP vs. SP |
| SOL | 3.67 | 11.00 | Z = -3.87 | 3.33 | 0.17 | Z = -0.33 | 2.33 | Z = -4.15 | Z = -3.42 | Z = -0.79 | Z = -4.45 |
| | (4.17) | (23.67) | **p<0.001** | (2.50) | (6.83) | p = 0.739 | (4.50) | **p<0.001** | **p = 0.001** | p = 0.424 | **p<0.001** |
| WASO | 16.00 | 10.50 | Z = -0.86 | 14.00 | 0.83 | Z = -3.23 | 13.00 | Z = -0.52 | Z = -0.60 | Z = -0.220 | Z = -2.01 |
| | (34.00) | (25.17) | p = 0.389 | (15.00) | (12.67) | **p = 0.001** | (23.00) | p = 0.605 | p = 0.546 | p = 0.830 | **p = 0.045** |
| NWE | 8.00 | 4.00 | Z = -4.39 | 7.00 | 3.00 | Z = -4.90 | 7.00 | Z = -0.75 | Z = -0.91 | Z = -0.210 | Z = -0.33 |
| | (13.00) | (4.00) | **p<0.001** | (7.00) | (5.00) | **p<0.001** | (7.50) | p = 0.451 | p = 0.361 | p = 0.835 | p = 0.739 |
| WASF | 1.33 | 5.00 | Z = -2.78 | 1.00 | 0.17 | Z = -0.52 | 0.83 | Z = -1.68 | Z = -1.678 | Z = -0.516 | Z = -2.83 |
| | (4.00) | (37.00) | **p = 0.005** | (2.33) | (4.00) | p = 0.603 | (1.33) | p = 0.094 | p = 0.093 | p = 0.613 | **p = 0.005** |
| Sleep% | 0.945 | 0.898 | Z = -2.86 | 0.952 | 0.967 | Z = -1.69 | 0.965 | Z = -1.39 | Z = -1.65 | Z = —0.120 | Z = -4.31 |
| | (0.084) | (0.164) | **p = 0.004** | (0.032) | (0.053) | p = 0.092 | (0.041) | p = 0.165 | p = 0.100 | p = 0.904 | **p<0.001** |
| No. of Markers | | 21.00 | | | 7.00 | | | | | | Z = 3.94 |
| | | (18.00) | | | (7.00) | | | | | | **p<0.001** |

Median values for ACT and M. Interquartile ranges in parentheses. Paired-sample Wilcoxon signed-Rank-Tests of simultaneous ACT and Marker measures for RP and SP in light grey. The four right-most columns show Z and p values for Mann-Whitney-U tests in ACT and M scores under different conditions in darker grey. Significant test results in bold letters.

RP = Pressing in response to vibration pulse, SP = Self-initiated Pressing, NP = No Pressing, M = Marker-based, ACT = Actigraphy, SOL = Sleep Onset Latency [min], WASO = Wake After Sleep Onset [min], NWE = Number of Wake Episodes, WASF = Wake After Sleep Offset [min], Sleep% = Sleep Percentage.

right-most column of Table 2. Marker measures for SOL, WASO and WASF were larger in RP than SP and smaller for Sleep%, indicating RP to be more sensitive to wakefulness than SP. However, equal NWEs indicate equal sensitivity of SP and RP for arousals, and no increase in arousals with the vibration pulse. Apparently, the vibration pulse did not wake up the subjects who accepted it. Equal numbers of wake epochs but higher values for SOL, WASO, and WASF mean that the duration of wakefulness was measured longer when responding to vibration. This might be either because the band gave better information about when to press again while lying awake, or because the vibrations kept subjects awake longer. Another possible underlying mechanism is that during wake-sleep transitions, subjects might be unable to self-initiate a behavioral response but still be able to respond to external stimuli. Different behavioral approaches of measuring sleep onset have been shown to lead to different measures for SOL during the sleep onset period [28].

## Simultaneous actigraphic vs. marker measures and agreement

Paired-sample Wilcoxon signed-rank tests (Table 2) showed that for RP, marker-based measures gave significantly higher results than actigraphy for SOL and WASF but lower ones for NWE and Sleep%. For SP, marker-based measures gave lower results for WASO and NWE, while SOL, WASF and Sleep% did not differ significantly. Spearman correlations between marker sleep parameters and actigraphy for both pressing conditions were generally not significant except for positive correlations for SOL in RP (r = 0.747, p<0.001), Sleep % in RP (r = 0.433, p = 0.015), and a weak correlation for WASO in SP (r = 0.240, p = 0.045). A better way of calculating agreement is illustrated by Bland-Altman plots (Fig 2) for SOL and Sleep% in the RP and SP conditions [29]. The two oblique lines represent the maximum mathematically possible deviation from the zero line. In all of the graphs, many points lie on or close to those lines, indicating a large discrepancy between activity and marker based sleep parameters. The *limits of agreement* are only approximate except for Sleep% in RP, because differences of measures were not normally distributed, and log-transformation did not normalize them [29].

For RP, the Bland-Altman plot shows that the mean difference in Sleep% was lower when measured by markers (mean difference = -5.2%, 95%-CI: -8.3 to -2.0%, t(30) = -3.379, p = 0.002). Limits of Agreement (with 95% CI) lay between -21.9% (-27.3 to -16.5%) and 11.5% (6.1% to 16.9%). For SOL in RP, we did not exclude the three outliers of longer periods of repetitive marker presses, since they were associated with low activity at the beginning of bedtime. At higher SOLs, measures spread more.

For SP, Bland-Altman plots of Sleep% show a clear divergence in both directions of disagreement, being greater with more deviation of the average from 1. SOL shows a similar pattern. As opposed to RP, there are two negative outliers resulting from not pressing within the 10 minute and 10 second time limit. For SOL values close to zero, a cluster of data points along the lower oblique line for maximum possible deviation means that actigraphically determined short SOLs were often shorter or zero when determined by markers.

Table 3 presents another index of agreement of measures, *epoch-by-epoch agreement*. For 31 RP nights, we evaluated 85,387 epochs or about 237 hours, and for 51 SP nights 138, 938 epochs, or about 386 hours. The agreement of markers and activity is shown in an error matrix for the two conditions separately. For the RP condition, overall agreement was 88.6% and for SP, 92.1%. Kappas, indexing congruent wake epochs and congruent sleep epochs, were moderate for RP and fair for SP. As expected, prevalence and bias adjusted Kappa (PABAK) values were generally higher. It is not surprising that for SP, despite only 1.5% wake agreement, PABAK showed almost perfect agreement, while for RP, agreement was lower.

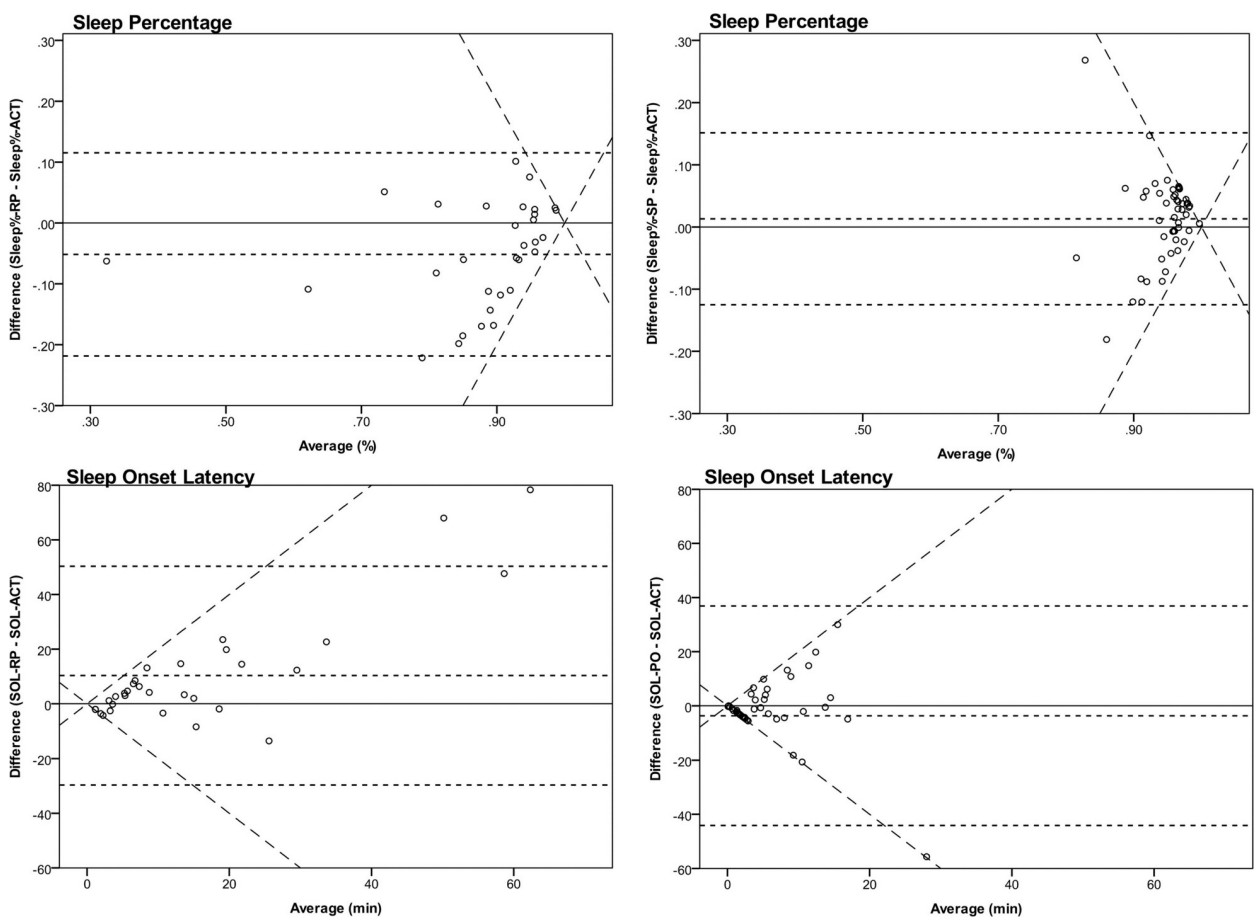

**Fig 2. Bland-Altman Plots for Sleep% and SOL in RP and SP.** Bland-Altman Plots for Sleep% and SOL determined by markers vs. ACT in RP and SP conditions. The dashed horizontal lines represent (from top to bottom) the upper 95% limit of agreement, the mean difference, and the lower 95% limit of agreement. The dashed oblique lines represent the maximum divergence of marker and ACT measures mathematically possible. In the lower right plot, one extreme value is not displayed: (63,-126). RP = Pressing in response to vibration pulse, SP = Self-initiated pressing, ACT = Actigraphy, Sleep% = Sleep Percentage, SOL = Sleep Onset Latency.

## Discussion

An acceptable reporting method should not be stressful or interfere with sleep, unlike RP. RP was rejected by 41% of our subjects, while SP was accepted by all, although subjects reported lower

**Table 3. Epoch-by-epoch agreement of M vs. ACT.**

| | Response to vibration (RP) | | | Self-initiated Pressing (SP) | | |
|---|---|---|---|---|---|---|
| | Wake scored by ACT | Sleep scored by ACT | Total | Wake scored by ACT | Sleep scored by ACT | Total |
| Wake scored by M | 6.1% | 8.2% | **14.4%** | 1.5% | 3.4% | **4.8%** |
| Sleep scored by M | 3.1% | 82.5% | **85.6%** | 4.5% | 90.7% | **95.2%** |
| **Total** | **9.3%** | **90.7%** | | **6.0%** | **94.0%** | |
| Overall agreement | 88.6% | | | 92.1% | | |
| Cohen's Kappa | 0.456 | | | 0.229 | | |
| PABAK | 0.773 | | | 0.843 | | |

Congruent wake and sleep epochs scored by M and ACT and total proportions. An epoch is defined as a 10-seconds time interval. M = Markers, ACT = Actigraphy, PABAK = Prevalence and bias adjusted Kappa.

sleep depth compared to NP. This rejection was greater than reported by Miller, Babler [13] who gave signals every 10 rather than 5 minutes, and may have given haptic signals of lower perceived intensity. The reason that SOL measured by markers was longer in RP than SP may be that the vibrations kept subjects awake. Log-reported adherence to instructions was that subjects "usually" pressed the marker, but pressing may have been less frequent for subjects worried about their sleep. Furthermore, accurately estimating SOL or WASO with SP depends on subjects knowing that they are awake and remembering to press every 5 to 10 minutes. When transitioning between wake and sleep, time estimation may be especially inaccurate. We had decided against asking subjects to respond more frequently, afraid that that would interfere more with their sleep.

A limitation of this study was the lack of simultaneous PSG recording. With PSG we would have had an additional measure of whether subjects were awake or too drowsy to follow the push instructions. However, even PSG classification of epochs as wake or sleep is less than certain [30]. EEG and behavioral definitions of being asleep give discrepant results. Pressing to auditory stimuli occurs not only during PSG wakefulness, but continues with decreasing prevalence in stage 1 and stage 2 sleep [9]. In one study, release of a dead man switch corresponded to EEG-measured SOL but not to EEG-measured WASO because of failure to press the switch when awake [9, 10]. Behavioral methods have a strong claim for face validity in that pressing a button at a certain point in time implies at least some level of wakefulness at that moment.

Pressing clearly gives information that supplements that given by actigraphy. Epoch-by-epoch analysis of RP detected wakefulness where ACT did not in 8.2% of epochs. SP detected less wakefulness but did in 3.4% of epochs where ACT did not. Actigraphy indicated higher NWE than did marker presses in RP and SP. In RP, markers indicated higher SOL and WASF than actigraphy, but in SP, both were shorter, although not significantly. In RP, WASO by markers and actigraphy did not differ significantly, but in SP WASO was longer by actigraphy. Shorter times could mean that actigraphy failed to correctly register subjects being awake when they were lying quietly, which is a prerequisite of falling asleep. Longer times mean either that actigraphy registered being awake when subjects actually were awake but failed to press the marker, or that actigraphic activity was above the wake threshold when subjects actually were asleep. According to actigraphy, pressing itself (SP) indicated no more overall wakefulness than NP for Sleep% or WASO. Higher actigraphic SOL in RP and SP can be explained either by motion artifacts due to marker pressing or because subjects were kept awake by the task.

In conclusion, pressing an event-marker adds important information to that provided by actigraphy, but at the cost of several drawbacks. First, any behavioral response potentially affects the sleep it is attempting to index. Second, when subjects perceive responding as disturbing to their sleep, they may reject it as unacceptable. By applying each condition twice, we found that SP was acceptable to all of our subjects, but that RP was not. However, SP did result in lower perceived sleep depth than the NP condition. Perhaps RP could be made more acceptable by adjusting vibration strength closer to individual thresholds, but in some subjects, vibration seems to evince an uncanny feeling, that could be bothersome at any level. Third, pressing tasks require a capacity to remember and follow instructions, which are likely to be impaired in two groups whose sleep is of interest, children and cognitively impaired adults and elderly. Thus, additional ways of supplementing actigraphy could be useful. Physiological possibilities include heart rate measured by photoplethysmography at the wrist e.g., Beattie, Oyang, et al. [31] or limited EEG data, e.g., Levendowski, Ferini-Strambi, et al. [32].

## Supporting information

**S1 File. Dataset.** Comprising PSQI, sleep diaries and all sleep data.
(XLSX)

**S2 File. Vibration Wristband setup.** Comprehensive technical information on our self-developed vibrating wristband.
(DOCX)

## Acknowledgments

The Department of Veterans Affairs Sierra-Pacific Mental Illness Research, Education, and Clinical Center (MIRECC) provided office space and technical supplies.

## Author Contributions

**Data curation:** Marius Keller.

**Formal analysis:** Marius Keller.

**Investigation:** Marius Keller.

**Methodology:** Marius Keller, Walton T. Roth, Katja Petrowski.

**Project administration:** Walton T. Roth.

**Resources:** Walton T. Roth, Katja Petrowski.

**Software:** Marius Keller.

**Supervision:** Walton T. Roth, Katja Petrowski.

**Validation:** Marius Keller, Walton T. Roth, Katja Petrowski.

**Visualization:** Marius Keller.

**Writing – original draft:** Marius Keller.

**Writing – review & editing:** Walton T. Roth, Katja Petrowski.

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
