## [Decision Letter · Decision Letter 0]

16 Apr 2020

PONE-D-20-03712

Supplementing sleep actigraphy with button pressing while awake

PLOS ONE

Dear Mr. Keller,

Thank you for submitting your manuscript to PLOS ONE. After careful consideration, we feel that it has merit but does not fully meet PLOS ONE’s publication criteria as it currently stands. Therefore, we invite you to submit a revised version of the manuscript that addresses the points raised during the review process.

Both reviewers have found your paper provides useful information and is worth publishing.

Reviewer 2 has some minor concerns and useful suggestions in order to clarify some points.

We would appreciate receiving your revised manuscript by May 31 2020 11:59PM. To enhance the reproducibility of your results, we recommend that if applicable you deposit your laboratory protocols in protocols.io, where a protocol can be assigned its own identifier (DOI) such that it can be cited independently in the future. For instructions see: http://journals.plos.org/plosone/s/submission-guidelines#loc-laboratory-protocols

We look forward to receiving your revised manuscript.

Kind regards,

Pierre Denise, Ph.D, M.D.

Academic Editor

PLOS ONE

Journal Requirements:

"This research was supported by the Department of Veterans Affairs Sierra-Pacific Mental Illness Research, Education, and Clinical Center (MIRECC)."

"The authors received no specific funding for this work."

Reviewers' comments:

Reviewer's Responses to Questions

**Comments to the Author**

1. Is the manuscript technically sound, and do the data support the conclusions?

Reviewer #1: Yes

Reviewer #2: Yes

2. Has the statistical analysis been performed appropriately and rigorously? 

Reviewer #1: Yes

Reviewer #2: Yes

3. Have the authors made all data underlying the findings in their manuscript fully available?

Reviewer #1: Yes

Reviewer #2: Yes

4. Is the manuscript presented in an intelligible fashion and written in standard English?

Reviewer #1: Yes

Reviewer #2: Yes

5. Review Comments to the Author

Reviewer #1: Comments on the paper entitled "SUPPLEMENTING SLEEP ACTIGRAPHY WITHBUTTON PRESSING WHILE AWAKE"

Comments

The study aims to investigate the impact, on sleep parameters obtained by actigraphy, of pressing an event-button when lying without any movement, but not in a drowsy state. Two system have been compared. In the first one, the button had to be pressed every 5 min when an haptic signal (vibrations throughout a wristband) is given. In the second one, instead of being reminded to press the button by a vibration, lying awake was to be signalled by pressing the marker about once every 5-10 min.

Results show that the first condition was unacceptable for about one out of two study participants while in the second one, is acceptable to many. The uncertainty on estimation of sleep onset while lying-awake is reduced compared to actigraphy without pressing a button. In epoch-by-epoch analysis, both technics detect more (3.4-8.2%) wakefulness than in actigraphy alone, with about two time more in the wristband stimulation.

This is an interesting paper, which have some relevance for interpreting the results obtained by actigraphy in sleep studies. However, sleep might be slightly affected by the instruction and this disturbance might be dependant of the type of patient recorded. One of the limit of the study is to have include any adult regardless of health status. The next step will be to compare the results obtained in young good sleepers (in which actigraphy have a very good correlation with polysomnography) and other population (olders, insomiacs, …).

Nonetheless, self-initiated button pressing to indicate being awake while lying in bed is acceptable to many, interferes little with sleep on average, and adds to the information given by actigraphy, this will certainly change the way this instrument is used.

Reviewer #2: As the authors point out, it has been known that autographs, while very useful in describing several sleep parameters, seriously misrepresent SOL and wakefulness during TIB. Supplementary, well validated behavioural measures could improve the accuracy of describing sleep and wakefulness during the night. The authors have chosen to use spontaneous and cued button presses for this purpose. While the paper provides a useful addition to the current literature, a number of suggestions follow which might clarify several points and perhaps extend the usefulness of the paper.

The introduction is well written and the goals of the study are clearly related to the experimental design. (An additional reference is needed on L 71.) Sound ethics and consent procedures were employed. The use of diverse participants increases the generalizability of the findings. Behavioural definitions are clearly described and the MotionLogger is a good choice for an actigraph. A programmable vibratory wrist band is a good way to deliver a cue for a behavioural response, but the physical parameters (Hz, amplitude) must be included for replicability. In such studies, the faintest cue possible should be used. Please state clearly that this study was conducted at home. L. 99 "usually consecutive" nights is imprecise. Elaborate.

In the Results, important information is omitted. The reader ought not to need to calculate overall compliance (131/180 nights = 73%: 31/60 for RP = 52%; 51/60 for SP = 85%; 49/60 for NP = 82%). The authors address the issue of the acceptability of the two button pressing conditions and found no significant differences between conditions, which is very fortunate, for they should have used a counter-balanced design in which there was an equal probability of any participant receiving any condition on any given night. In general, the results are thoughtfully analyzed and well presented.

Under Actigraphic Measures (L 230-234) the description of differences in SOL is not clearly presented. It is important to fully consider why SOL might be longer in the cued response condition (RP) than in the non-cued conditions. Authors need to go back to the definitions of their three conditions to fully explain their data.

Marker measures: L235-43 Possible explanations - or that the marker measures index different points on the W/S/W continuum. This comment also applies to the next section on Simultaneous actigraphic vs marker measures. Also, in L 246, it is stated that RP underestimated SOL, but data from Table 2 shows the opposite - which is more in line with expectations. Please check the remainder of Table 2 against interpretations. Also, the section is written as if the autograph is the Gold Standard for measuring sleep. I thought that the author's premise was that additional behavioural measures might fine-tune sleep measurement. Please be consistent with your own Introduction. Please re-write indicating which measures give the shortest to longest estimates of SOL. If you want to compare against a Gold Standard, repeat the study including polysomnography, focussing on EEG changes and look to see what percentage of the variance between EEG and behavioural measures can be accounted for with various weightings of behavioural measures. This also applies to interpretation of other measures.

L 285 - The fact that the SP condition shows only moderate agreement suggests that this measure may acc our for additional variance when describing W and S and therefore may indeed be a useful metric.

The Discussion is well written and argued.

Further to L.315, normally, lying still is a prerequisite for falling asleep and remaining so but is not convincing evidence of same.

In sum, after minor revisions, the paper will make a useful addition to the literature and uncued responses can easily add to the accuracy of the autograph in describing sleep and wakefulness.

6. PLOS authors have the option to publish the peer review history of their article (what does this mean?). If published, this will include your full peer review and any attached files.

Reviewer #1: No

Reviewer #2: No

---

## [Author Response · Author response to Decision Letter 0]

16 May 2020

See rebuttal letter ('Response to Reviewers').

---

## [Editor Report · Decision Letter 1]

19 May 2020

Supplementing sleep actigraphy with button pressing while awake

PONE-D-20-03712R1

Dear Dr. Keller,

We are pleased to inform you that your manuscript has been judged scientifically suitable for publication and will be formally accepted for publication once it complies with all outstanding technical requirements.

With kind regards,

Pierre Denise, Ph.D, M.D.

Academic Editor

PLOS ONE
---

## [Editor Report · Acceptance letter]

4 Jun 2020

PONE-D-20-03712R1 

Supplementing sleep actigraphy with button pressing while awake 

Dear Dr. Keller:

I'm pleased to inform you that your manuscript has been deemed suitable for publication in PLOS ONE. Congratulations! Your manuscript is now with our production department. 

Kind regards, 

on behalf of

Pr. Pierre Denise 

Academic Editor

PLOS ONE